# The STEP Program—A Qualitative Study of the Supportive Therapeutic Excursion Program and Its Effect on Enabling Parental Self-Efficacy and Connectedness after the Stress-Experience of the NICU

**DOI:** 10.3390/children9050591

**Published:** 2022-04-22

**Authors:** Makini McGuire-Brown, Rudaina Banihani, Jo Watson, Eugene Ng, Colleen Rocha, Laura Borges, Paige Terrien Church

**Affiliations:** 1Schulich School of Business, York University, North York, ON M3J 1P3, Canada; makinidm@yorku.ca; 2DAN Women and Babies Program, Sunnybrook Health Sciences Centre, Toronto, ON M4N 3M5, Canada; rudaina.banihani@sunnybrook.ca (R.B.); jo.watson@sunnybrook.ca (J.W.); eugene.ng@sunnybrook.ca (E.N.); colleen.rocha@sunnybrook.ca (C.R.); laura.borges@sunnybrook.ca (L.B.)

**Keywords:** preterm birth, NICU, parents, attachment, parental mental health

## Abstract

As a child is born, so are the parents with new roles assumed. Attachment is secured, and parents quickly become the experts in their child. With preterm birth, however, this quiet and essential period of discovery is interrupted. It is described as a traumatic and stressful experience. Programs that aim to address this focus on education and support. This qualitative study explores the effect on parents and staff of a novel and simple intervention of parents walking their baby in a stroller (Supportive Therapeutic Excursion Program, STEP). This study used a qualitative methodology that included 20 semi-structured interviews for content saturation to explore the experiences of parents and staff participating in the STEP program. We found an overall positive experience reported by both parents and nursing staff with STEP. Analysis revealed that as autonomy is fostered with parent education around transition to home, STEP brought a normalcy that was reported to bolster participating parents’ self-efficacy and connectedness, two important elements for recovery from stressful situations. The potential impact of STEP can be far reaching because, although future research is needed, these data suggest that inexpensive programs that promote normalcy may enable self-efficacy and connectedness prior to hospital discharge.

## 1. Introduction

Pre-term birth is a sentinel moment for a family. As the child is born, so are the parents with new roles assumed. It is also described as traumatic and stressful for the parents [1,2]. Effects have been researched in the acute phase [3,4] as well as the consequential long-term effects [5]. Given these identified consequences with higher rates of anxiety and depression in parents, higher levels of parenting stress and poorer family functioning [5], programs have been developed to decrease the traumatic effects of preterm birth on parents and therefore children. These programs are numerous but strong empirical evidence exists for Kangaroo Care (KC), Mother Infant Transaction Program (MITP), Infant Health and Development Program (IHDP) [6] and the Family Integrated Care (FICare) program [7]. The targeted components in these programs include parent education, parent support and infant therapy/support [6,8]. Systematic reviews of these programs demonstrate benefits in infant outcomes, mother-infant dyadic outcomes and parental outcomes [6]. Positive effects on anxiety, depression, and self-efficacy have been demonstrated [8]. In the FICare trial, there was improvement in daily baby weights, frequency of exclusive breastfeeding at discharge as well as lower stress and anxiety scores [7].

To date, programs for parents have focused on parental education and support with positive results but fall short on enabling self-efficacy and connectedness through normalcy. Self-efficacy and connectedness [9] are two key elements of the gold standard therapeutic approach, Cognitive Behavioral Therapy (CBT), for Acute Stress Disorder (ASD) and Post-Traumatic Stress Disorder (PTSD). Both ASD and PTSD are common in parents of preterm infants [10]. Self-efficacy is defined by the American Psychological Association (APA) as a subjective perception of one’s capability to perform in a given setting or to attain desired results [11]. Connectedness, the sense of community and belonging [12], is important to cultivate in “premature parents” [3], who feel detached from “regular parents”. It is worth exploring then, if programs designed to ameliorate the psychological distress of premature birth should include a focus on typical parenting activities that may further enable a sense of self-efficacy and foster connectedness. Similar to how CBT for ASD aids in decreasing the chances of progressing to PTSD [9], enabling self-efficacy and connectedness during the NICU journey may offset the long-term effects previously mentioned [5].

A limitation to the facilitation of autonomy and restoration of self-efficacy is the lack of moments of time truly alone with the preterm infant, those windows of time for a parent to know their child and become the experts [3]. There is always an ‘expert’ outside the curtain, down the hall, or with the push of a button. This phenomenon contributes to the seismic nature of discharge, when that supportive presence is withdrawn, leaving parents with fledging skills but no certainty of capacity.

The Supportive Therapeutic Excursion Program (STEP) is a program which enabled parents to take their baby for a walk in a stroller on hospital grounds. The purpose of this program was to provide independent time to practice parenting and foster a sense of self-efficacy and connectedness, starting with walks around the NICU and advancing to walks on the hospital property. Anecdotally, parents described these walks as very positive experiences, some becoming tearful as they stepped into the sunshine with their baby for the first time.

The purpose of this qualitative study was to formally explore parental and staff perception of STEP. The objective was to learn about barriers, parental perception and experiences as well as to explore staff experiences, barriers and perceptions of the walks. We hypothesized that the walks would be positive for parents and that parents would report that the walks bolstered parental self-efficacy and connectedness. We also hypothesized that the stroller walks may be met with mixed perception from staff, given the added work of supporting parents as they transition from a fixed monitor to a mobile monitor and become accustomed to growing parental independence.

## 2. Materials and Methods

### 2.1. Method

Due to the lack of prior research with this intervention, we used a qualitative methodology that included semi-structured interviews to explore the experiences of parents and staff participating in the STEP.

At a tertiary, urban NICU, the intervention of therapeutic stroller walks, called STEP, was initiated in 2016 with parents of extreme preterm babies or babies with significantly complicated medical courses preventing transfer to a level II nursery. The babies were in a convalescent phase of their medical course and no longer acutely unstable. With STEP, parents were enabled to take their infant for a walk in a stroller. A policy for the walks was developed with input from hospital infection prevention, risk management, and legal expertise (see Appendix A).

Infants deemed eligible for STEP included those that were no longer on invasive ventilation, over 35 weeks corrected gestational age, and no longer having clinically significant apneic or bradycardia events. The stroller needed a lie flat pram component to provide ease of posture for the infant and direct visual access for the parent. Hospital grade sanitizer was provided. For those infants on supplemental oxygen, they were transitioned to a portable oxygen tank. Monitoring for heart rate, respiratory rate, and oxygen saturation continued, on a portable monitor. The walks progressed from around the NICU corridors to quiet but off the unit areas within the hospital and lastly, weather permitting, outside to hospital green spaces. Parents were advised on safety steps such as acute management, infection control policies, and a medical order was written to accommodate the walk activity.

### 2.2. Sampling

We used purposeful sampling and contacted families and nurses that participated in STEP throughout the years 2016–2020. All parents and nurses were willing to participate and give consent, there were no rejections. Sampling was carried out to saturation [13], defined as having enough data so that the researchers can produce comprehensive and convincing theory. Hennink, Kaiser and Marconi [14] also categorize meaning saturation as that necessary to “understand it all”; further data collection not adding to the content. After 10 interviews with parents and 10 interviews with nursing staff, the themes identified were consistent and further interviews were not generating novel thematic content. Morse [13], also suggests that the more cohesive the sample, the faster saturation is reached. As the participants in the STEP program experienced the common experience of preterm birth with a consistent NICU environment at the same hospital, and with the same NICU staff, there was a degree of cohesion. It is understandable then that saturation could be reached after 20 interviews.

Recruitment of eligible parents was via email and phone invitations from a study coordinator, who was not involved with STEP and had no prior knowledge of the patient families. In addition, social media invitations were posted on the clinic Facebook™ site and Instagram™ sites. Recruitment of NICU staff was via the weekly staff newsletter. Posters with the study information and contact were placed in the NICU.

### 2.3. Data Collection

The study was approved by the Research Ethics Board (#4948). After informed consent, a mutually convenient Zoom™ videoconferencing interview with the study coordinator and participant was established. Other authors who had prior knowledge of the families were not in attendance to prevent the introduction of acquiescence or social desirability bias. The interview was recorded for transcription purposes and data stored on locked software.

Semi-structured interviews for parents and staff were piloted on staff and parent liaisons and we then conducted interviews with 10 patients and 10 nursing staff from September to December 2021 at which point saturation was achieved (See Table 1 for participant characteristics).

Essential components of the interview guide for parents included: basic medical information about their child to provide a better understanding of the experiences to date for families; parents’ thoughts on the walks, to explore if they found them positive, empowering, or frightening and unsettling; the impact on their sense of parental self-efficacy, to explore if these walks allowed them to feel more competent as a parent; and barriers and facilitators, to explore operational aspects that helped or interfered with the ability to take walks. (See Appendix B for full interview guide.)

Essential components of the interview guide for staff included: their role and years of experience; their impression of the impact of walks as beneficial or detrimental to parental capacity; the impact on their workload; and facilitators and barriers, to explore operational aspects that helped or interfered with the ability to take walks. (See Appendix B for full interview guide.)

As is recommended in qualitative research and when using the semi-structured interview instrument [15], the process was iterative and additional probing and follow up questions were asked to fully understand the lived experience of the participants.

### 2.4. Data Processing and Analysis

Data was transcribed and corrected for any errors in transcription following each interview by the interviewer. The transcripts were then de-identified before distributing for coding, to avoid contamination from some researchers’ prior knowledge of the participants. All transcripts were coded separately by each author and then codes were compared and discussed to determine congruity. Coding was carried out using the tabula geminus approach [16] that links emic, in vivo codes, taken from the participants’ language, to etic codes that include theory applicable to the study. Data from the parent participants were analyzed as a unit. They were interviewed together and there was very little difference in perspective between the mother and father of each child, although, this is not something that this study explored in depth. From the data, a deeper understanding of the effect of STEP on the patients and staff was obtained and formulated into theory.

## 3. Results

Participation in STEP was reported to be a positive experience for parents from both parents and nursing staff. Barriers and improvements were identified. The themes identified included: the NICU as an intensely stressful experience for parents; trust transfer, necessary process elements for staff, and benefits for parents; normalcy of the experience and parents’ report of emerging autonomy, self-efficacy and connectedness; and observed benefits to mental health. Analysis revealed an intimate and integrated relationship in which trust transfer births autonomy, autonomy is the gateway for STEP and STEP was reported to bolster self-efficacy and restore connectedness, two important elements for recovery from stressful situations. These findings are detailed below and illustrated in Figure 1.

### 3.1. NICU as an Intensely Stressful Experience

Parents consistently identified the NICU as stressful mentally, emotionally, and physically. During this time, they experience intensely negative emotions.


*“It was the worst time; it was horrible…. NICU changes you. And like, I went from being a realist to being a pessimist, like, very much.”*
Participant 5 (parent)


*“It was honestly the toughest time of our lives I guess”*
Participant 1 (parent)


*“Stressful, very stressful. I mean, everyone was helpful, and it wasn’t anything to do with the care, just being there was stressful.”*
Participant 13 (parent)


*“I had some already latent mental health concerns and then the NICU just brought that up.”*
Participant 12 (parent)

Parents themselves felt helpless but it was also evident that a big part of the experience was trust in the healthcare team. Handing over this trust to the healthcare team was a coping strategy used.


*“One of our goals was to just completely defer our trust to the doctors and to not sort of be marked down by some of the panic and the fear.”*
Participant 6 (parent)


*“Just because like we trusted the doctor so much…so I had full faith that if they’re ready, I’m definitely ready.”*
Participant 1 (parent)

### 3.2. At the Right Time

There appeared to be a time when the trust was transferred the other way; the staff placed trust in the parent to begin taking greater control in the care of their child. Influencing factors included infant factors, parent factors or legal/policy factors. Staff members were reluctant to transfer trust if an infant was deemed too ill or if the parents had not demonstrated skills needed to go for a walk. This was particularly salient for STEP in which the parent was allowed to leave the confinement and safety of the NICU. Legal/policy factors be considered as this may affect the trust transfer process. While the sentiment was generally positive toward the introduction of STEP, participating nurses identified criteria that in their mind would be suitable for determining eligibility to participate in STEP, such as no bradycardic or tachypnoeic spells that require intervention, no desaturations that require intervention and parents trained in CPR. It was important to nurses that they have policies that reflect this “right time” otherwise they feared legal ramifications. Finally, teamwork was identified as a necessity and therefore a possible limitation if staff coverage was inadequate.

#### 3.2.1. On the Institution of STEP


*“Doing something that a normal family would get to do, like going for a walk is just probably as normal as breathing. Right? For the NICU parents, they’ve been deprived of a lot of things. Yeah, they likely haven’t had the baby shower they’ve maybe not ever had, they haven’t had the prenatal education classes because they already had a baby, the baby. You know, they haven’t had, the mother…. hasn’t had that experience of having a huge belly and everybody asking… so there’s all these things that are normal in the walk of pregnancy that are our moms don’t get. So trying to bring them these pieces of normal; I think it’s so important.”*
Participant 7 (staff)

#### 3.2.2. On Eligibility Factors


*“I mean, they have to be for sure stable, physically stable, from a cardiac respiratory point of view, or the one that has the odd desaturation or bradycardia, the parents know how to respond. And we’ve been through this scenario many times, so it’s an older baby with reflux. And every once in a while, for sure they have a bit of reflux. And, you know, they do some breath holding, they dipped down quickly, but right back up, and the parents know how to intervene. And this is something that they’ll be dealing with at home, then I don’t have a problem with that. But if I was in the same scenario, if it was a parent who couldn’t demonstrate that they knew what to do, that I’m not okay with that.”*
Participant 2 (staff)

#### 3.2.3. On Legal/Policy Factors


*“The stroller walks, the big concern as well, like liability. And what if you know, something happens when they’re out with their baby? I mean, you’re the one that’s caring for their baby. But if you don’t go with them, how much of that is your responsibility? And I think that’s where the resistance comes from. I think a lot of the nurses that are concerned, feel like should something happen when the moms go out with the baby, that’s our, our license, essentially, by sending them out there. So I think that is the huge, the huge part of the resistance.”*
Participant 4 (staff)

#### 3.2.4. On Teamwork and Staffing


*“It just requires me to reorganise maybe my time a little, you know, if I’m, if I’m truly busy, I’ll say to them, you know, right now, I’m sorry, I can’t take you outside right now, if you want to go for a walk in the hallways feel free to do that. This is the time that I will be available, or I’ll negotiate with a colleague, maybe who’s, who’s free to go and take them out. So it’s just a little bit of juggling, and it’s what we do all the time in the NICU anyways.”*
Participant 20 (staff)


*“There are cons to the nurses too. If the nurse has got a really difficult assignment, like if you’ve got a baby that’s a little bit sicker with a baby that Mom wants to take it out for a stroll. I mean, it becomes like, Okay, well, [not now] because I’ve got this baby getting a blood transfusion or, we’re short staffed. And if we’re really short staffed, and if we’re really, you know, it shouldn’t matter, but it does matter. Right? And so, you know, sometimes it becomes a negative having to [send this] baby out for a walk, because, because you’ve got so many things to do. So you know, you’re encouraging the family unit and developmentally and, and all the things that go with it as, as sending out positive parents and positive vibes. But as I say that that one negative is the acuity of the unit.”*
Participant 9 (staff)

### 3.3. Trust Transfer

Parents who had placed their ability to cope in the trust of the care team expressed feelings of great autonomy when the team transferred trust back to them.


*“Okay, well, you’re the parent, you’re the mom, you know, so we’re going to transition you into more making those decisions. And I felt like that was very, very helpful for me because, like I was quite freaked out. Like I didn’t know, I didn’t know how to be a parent, right? Like, I actually felt really supported. And also it was a bit of a necessary push for me to take ownership over, like making some decisions I hadn’t made previously really.”*
Participant 10 (parent)


*“I think it’s definitely good for parents to participate [in taking on more decisions], just because it gets them into that next level. When I was with and in the NICU, like, I want to know what’s the next the next step and, and getting him home.”*
Participant 13 (parent)

### 3.4. Autonomy, Self-Efficacy and Connectedness

Parents articulated emerging autonomy as their baby convalesced and STEP provided the opportunity to exercise it. The “normalcy” of the activity was reported by parents to affirm them as parents, a parent who “can do it” (self-efficacy) and connectedness to “parent-like activity”.


*“I was [now] in charge of my child. And that it was up to me to decide how to be safe or not safe. And to give myself a little bit of self-confidence, and I think that’s important for moms, especially when you’re getting closer to discharge, it’s the lack of self-confidence and knowledge in your own capabilities of being a mother because it’s stripped from you right at the door.”*
Participant 12 (parent)


*“I have had a baby who’s had a small episode of spitting up or vomiting while they went out for a walk. But the parents, the parents managed it very appropriately, they turned the baby on his side, they wiped his mouth. They came back. And obviously they reported to me what had happened, but they weren’t, they weren’t more anxious. They weren’t, they weren’t freaked out about it, they, they handled it really well. I’ve never had a baby who had an actual spell, sometimes the monitor will go off because the baby’s kicking. But by that time, usually I find the parents because they have been present. They understand they look at their baby, they see their baby is pink and breathing. And they realise that the monitor is not picking up the reading properly. So that it’s not a concern with the baby. It’s really just a problem with the, with the connection.”*
Participant 20 (staff)


*“I remember putting on my jacket and tucking to the wind and like walking onto the elevator and it felt so normal. You don’t really think about when you’re in the NICU because you are so isolated… and everybody there is in the same position like physical and mental space. I also didn’t socialise that much because you just invariably end up talking about, you compare your babies and did they get this steroid and what how much food are they taking and how much weight did they gain and it feels like even though it starts off wholesome it always kind of ends in a weird I never really felt like I liked it. So anyway, it felt so nice just be able to, like I was just a regular mom walking through the halls with her baby.”*
Participant 6 (parent)


*“It feels good that the medical professionals are putting their faith and confidence in you as a parent to be alone with your kid…. I went up there and I got a coffee and that was that was pretty great. Like [with] the stroller finally I could just sit with her. She’d like sort of be sleeping or just kind of relaxing and I could finally just have a coffee. And then I was like walking the hallway. And I ran into some of the other moms that I knew. And so that was really nice to be able to be like, ‘here’s my kid. She’s doing great’. And one of the other moms, I think, a few, like, twice, maybe three times, she happened to be on a stroller walk with her baby at the same time as me as well. So we kind of like walked a little bit together. And that was pretty great. That felt pretty, like normal, you know, mom behaviour.”*
Participant 10 (parent)

### 3.5. Mental Health

The benefits of this on the mental health of the parents were expressed by both the parents and the nurses.


*“It was nice. It was really nice because it gave us a break from the old little room, just like to breathe outside and just not constantly look at the numbers and have you fixated with the issues that she could have.”*
Participant 1 (parent)


*“I think it was like 109 days, I kind of went a bit crazy. And I was like, I’m not a prisoner in here and I just need to get outside. So this led up to us going outside for a walk. I was like, ‘I just need to go outside like I can’t be in this. I’ve been in here since June and it is now Thanksgiving and I just need to go outside’.”*
Participant 6 (parent)


*“I think stroller walks and especially being able to go outside with your infant, even if it’s just around the grounds; it gives them a chance to sort of improve their mental health and then ground them a little bit more with their, with their child.”*
Participant 4 (staff)

## 4. Discussion

This study explored parental and staff perception of a simple everyday parent activity, walking with their infant in a stroller. We identified themes of the stress of the NICU experience; trust transfer as coping and then autonomy building; and normalcy as an opportunity for self-efficacy and connectedness. This is complementary with prior research while making additional contributions. Our participants match those in other studies who describe the intense stress associated with pre-term birth and the NICU experience [17]. Programs involving support and education continue to be important [6] and, in this model, help in two areas. Firstly, programs with supportive and educational elements introduce autonomy, and secondly, they contribute toward the “right time” for trust transfer. Staff concurred that in order to be eligible to take the next STEP, the parents needed to demonstrate competence. Competence can only be achieved through programs with educational elements.

This study identifies the transfer of trust as a major element of the NICU journey that contributes to parent autonomy. At the beginning of the journey, parents have little choice but to trust the care team with their child. This trust is the gateway to autonomy because when the care team then transfers the trust over to the parent to begin having a greater role in the child’s care, the parents trust that the team is right and believe they can do it because the trusted team says the parent can. This raises interesting questions as to the duty of the care team to not only ensure evidence-based medicine but also with a duty to build rapport and trust with the patient family. Future research can explore the significance of this trust relationship and the effect of the degree of it or lack thereof on the autonomy built by parents.

Care must also be taken to build the right environment that allows for the trust transfer to the parents. This environment should include programs in place that would offer education and support the emergence of parental competency and unit policies supporting increasing autonomy, adequate staffing, and collaborative teamwork. An environment saturated with these elements is ideal to supporting the trust transfer from care team to parents.

What we uncovered is that of the theme of parent autonomy as a step toward efficacy. Autonomy denotes a feeling that you are now making your own choices [11] but self-efficacy goes beyond this. Self-efficacy denotes that you believe in your capability to make the decisions and perform the task and succeed [11]. Participation in STEP was reported to bring a normalcy that appeared to convince parents that they could be a parent. Parents saw themselves doing a task that normal parents do, they were on their own with their baby being the baby’s parent. They engaged in “normal” activities while with their baby and settled into their role as parent. This experience, backed by the foundation of education and support was reported by parents to instill a sense of self-efficacy and connected them to the world of parenting. This grounding has been reported to be essential in trauma recovery [9,18].

There are limitations to this work, mainly that being that it is limited to qualitative data. To date, there is no literature evaluating the intervention of therapeutic stroller walks, thereby limiting the ability to test the impact in a more rigorous way. The goal of this study was to identify possible themes and promote more research on this simple, inexpensive and novel intervention. The other limitation was the level of experience amongst staff who participated in the interviews, and in STEP. The participating nurses had at least 10 years of work experience which may impact on the generalizability of the perceptions expressed.

## 5. Conclusions

True to the strength of qualitative research, this focused sample provided an understanding of how a simple act of normalcy with STEP was reported by parents to contribute to parent self-efficacy and connectedness. While staff were cautious in their response, the staff did appreciate the benefit that they observed for parents and relied upon policy development. This study’s theoretical contribution of using trust to birth autonomy, and normalcy to enable self-efficacy and connectedness may be applicable to other families in the NICU since the stress-experience of NICU is universal. The potential impact of STEP needs to be explored further but this qualitative data suggests that fostering opportunities for parents to independently perform everyday parent activities was positive for parents and feasible for staff.

## Figures and Tables

**Figure 1 children-09-00591-f001:**
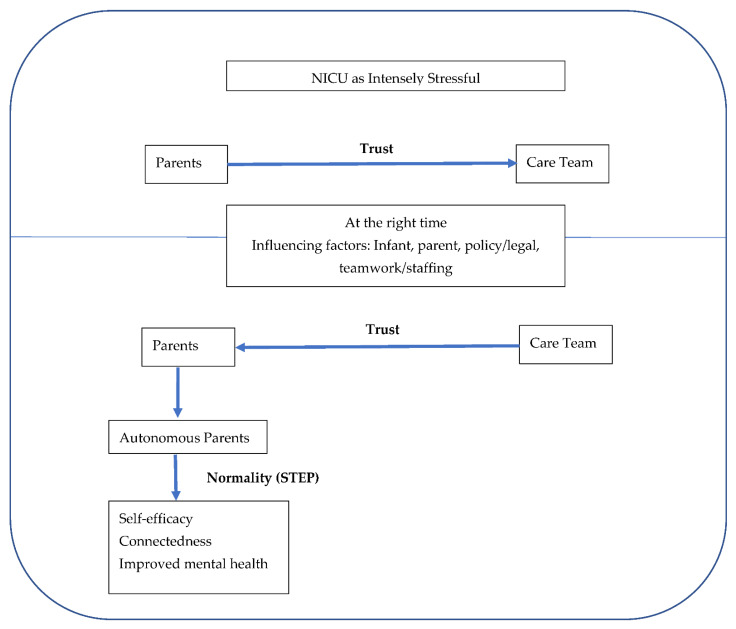
Trust and Normality enabling Self-efficacy and Connectedness in NICU parents.

**Table 1 children-09-00591-t001:** Interviewee Characteristics.

Participant Number	Role	Characteristics	Time between Discharge and Interview (Months)
1	Parent	GA at birth 24 + 6	10
3	Parent	GA at birth 23 + 6	21
5	Parent	GA at birth 30 + 5	22
6	Parent	GA at birth 29	58
10	Parent	GA at birth 25 + 1	19
11	Parent	GA at birth 25	29
12	Parent	GA at birth 24	41
13	Parent	GA at birth 25 + 5	23
14	Parent	GA at birth 23 + 5	10
15	Parent	GA at birth 24	43
2	Staff	Beside Nurse; 33 years of experience	N/A
4	Staff	Beside Nurse; 18 years of experience	N/A
7	Staff	Nurse Practitioner; 34 years of experience	N/A
8	Staff	Nurse Practitioner; 18 years of experience	N/A
9	Staff	Bedside Nurse; 30 years of experience	N/A
16	Staff	NICU Team Leader-7 years; Bedside nurse-17 years	N/A
17	Staff	NICU Team Leader; 31 years of experience	N/A
18	Staff	Bedside Nurse; 10 years of experience	N/A
19	Staff	Bedside Nurse; 20 years of experience	N/A
20	Staff	Bedside Nurse; 22 years of experience	N/A

## Data Availability

The data presented in this study are available on request from the corresponding author. The data are not publicly available due to the nature of the study.

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
