# Peer review of "The STEP Program—A Qualitative Study of the Supportive Therapeutic Excursion Program and Its Effect on Enabling Parental Self-Efficacy and Connectedness after the Stress-Experience of the NICU"

_children, 2022, doi:10.3390/children9050591_

Round 1

Reviewer 1 Report

This is a qualitative study of STEP – which is a local program facilitating parents to walk their preterm infant in a stroller in the hospital prior to discharge, a normalizing activity meant to foster ease of transition home. It’s a neat intervention and along with the qualitative study itself, the paper describes the program including the policy at the hospital, which can serve as a guide for others seeking to implement a similar program. The paper is well written overall and nicely presented. The background is well described and the literature review on related interventions is nicely done. The methods are well described and seem appropriate. The results are well presented including positive aspects as well as some of the challenges found through the interviews.

The stroller set-up is well described. Is there any chance a figure or photo could be provided to show the layout?

Figure 1 is generally nice but in final version could be improved in terms of font (appears somewhat faint), some inconsistency in box outlines (like the “Normality (STEP)” box left border), some words being cut off (bottom of ‘teamwork/staffing’).

Author Response

Thank you for these important comments.  Please see below for our replies.

The stroller set-up is well described. Is there any chance a figure or photo could be provided to show the layout?

We  did not have a specific image as we are not endorsing a specific brand but rather the lie flat pram style which we described in the methods.  If a drawing is felt to be needed, we can try to upload one. 

Figure 1 is generally nice but in final version could be improved in terms of font (appears somewhat faint), some inconsistency in box outlines (like the “Normality (STEP)” box left border), some words being cut off (bottom of ‘teamwork/staffing’).

Thank you for this important observation.  We have tried to edit it and it appears readable on our version.  Please let us know if that gets lost in the upload process.

Many thanks for your input. It has made our draft richer.

Reviewer 2 Report

i applaud the efforts made, and the approach taken to report on these aspects. I have seen these practices in different units where i was active as neonatologist, but there is for sure value in reporting on the parent perspective. I only have minor suggestions, to further improve this. 

'parent selection': do you have any idea on consent/no consent, and were parents data analysed as a unity, or do you have differences in perspectives between the mother and the partner (most commonly the father) involved ?

can you add the information on the timing of the interview versus the discharge of the infant ?  

Discussion: i value the reflections on the fact that this STEP approach can only be considered as part of a broader program and parent support approach.

Author Response

Thank you for these important comments and questions.  Please see below for our replies.

  1. 'parent selection': do you have any idea on consent/no consent, and were parents data analysed as a unity, or do you have differences in perspectives between the mother and the partner (most commonly the father) involved ?

Thank you for this question.  We have edited the manuscript to clarify that there no refusals from parents to participate.  Parents were unanimously engaged with being interviewed.  Staff that responded to advertisement also unanimously consented.  The data for the parents was analyzed as a unit.  It is an excellent idea to have analyzed by differing perspectives of mothers vs fathers but we did not collect this specifically.  That being said, there was little variation in the responses from parents.

2. can you add the information on the timing of the interview versus the discharge of the infant ?  

Thank you for this recommendation. We have updated the table accordingly.

3. Discussion: i value the reflections on the fact that this STEP approach can only be considered as part of a broader program and parent support approach.

Thank you so much!  We thought that this was an important point.

Thank you again for these important recommendations and questions.  You have made our paper stronger.